# The Coupling Coordination between Digital Economy and Industrial Green High-Quality Development: Spatio-Temporal Characteristics, Differences and Convergence

**Li Liu** [1,*] **, Tingting Gu** [2] **and Hao Wang** [1,*]

1    School of Business, Jinling Institute of Technology, Nanjing 211169, China
2    School of Humanities, Jinling Institute of Technology, Nanjing 211169, China
*    Correspondence: liuli1@jit.edu.cn (L.L.); wanghao@jit.edu.cn (H.W.)

**Abstract:** The coupling coordination between the digital economy and industrial green high-quality development has become an inevitable choice to promote economic high-quality development in China. This paper conducts an empirical analysis of the coupling coordination degree between the digital economy and industrial green high-quality development via the entropy evaluation method, coupling coordination model, Dagum Gini coefficient and its decomposition, and $\beta$ spatial convergence model, based on the spatial data of 31 provinces in China for a time period ranging from 2008 to 2020. The major research findings of this study are as follows: (1) the coupling coordination between the digital economy and industrial green high-quality development represents a whole steady-upward trend in China, with large regional differences. (2) There is a gradual decrease in the overall spatial difference of the coupling coordination, with the largest intra-regional difference in the eastern region, and significant inter-regional differences in the east-west, east-northeast, and east-central regions. Moreover, the hypervariable density serves as the major source of the regional differences. (3) There exists $\beta$ Convergence for coupling coordination degrees of the whole country and the four regions. However, the spatial effects are different in different regions due to different influencing factors. Therefore, sufficient attention should be paid to the dynamic trend, the difference, and the imbalance of the coupling coordination degree between the digital economy and industrial green high-quality development. The research is of great significance for accurately implementing policies according to different levels of local resource endowments and economic development, and narrowing the regional differences of the coupling coordination between the digital economy and industrial green high-quality development.

**Keywords:** digital economy; industrial green high-quality development; coupling coordination; regional differences

## 1. Introduction

Presently, China's economy is at a stage of high-quality development. Consequently, green development has become an inevitable choice. In addition, the industrial sector serves as the major carrier of economic development. It provides an effective guarantee for economic transformation and upgrading and high-quality development by accelerating the transformation of industries into factor-intensive, resource-conserving, and ecological protection-based industries; thus realizing the industrial green high-quality development. The digital- and real economy have been deeply integrated, owing to the popularization and application of new IT solutions supported by artificial intelligence such as 5G, IOTs, and big data. Moreover, new models, scenarios, and different forms of industrial development, which bring new opportunities for industrial green development, have emerged. Industrial green high-quality development represents a development model that human beings utilize human beings coordinate with the ecology. Meanwhile, the digital economy

represents a critical driving force for industrial green high-quality development. The digital economy is the driving path. Both need to be coordinated and coupled. How is the coupling coordination between China's digital economy and industrial green high-quality development? What are the characteristics? How about regional differences? These all need in-depth analysis. This paper constructs the index system of the digital economy and industrial green high-quality development, uses the coupling coordination model to measure the spatio-temporal characteristics of the coupling coordination level, and analyzes the spatial difference by the Dagum Gini coefficient and its decomposition, and the convergence by the $\beta$ spatial convergence model. These studies help us understand its change characteristics and evolution trends, improve the coupling coordination level between the digital economy and industrial green high-quality development, and promote China's high-quality economic development.

Tapscott (1994) has primarily proposed the concept of the "digital economy", and he explores the micro impacts and relevant business models. He indicated that the digital economy is "a new economy based on human intelligence and networking driven by interactive multimedia, the information superhighway and the Internet" [1]. Many scholars have made in-depth research on the meaning, function, characteristics, measurement and risk control of digital economy conduct in-depth studies on the meaning, function, characteristics, and risk control measures of the digital economy [2–7]. The studies on the industrial sector are reflected in two aspects, namely: digital industrialization and the digital transformation of industries. Digital technologies expand new models and formats; thereby supporting new industries and business forms [8]. Digital infrastructure is an important carrier for industrial development [9,10]. Furthermore, a digital industrial cluster promotes the formation of a high-quality digital industry chain. It also improves the efficiency of industrial innovation and the level of industrial collaboration through technology spillover, diffusion, and platform effects [11–13]. The digital transformation in different industries including agriculture, energy, manufacturing, and other fields at the regional and national level can be studied by integrating innovation between digital technology and traditional technology and making use of important scenes of traditional industries. As a result, the production modes, processes, and management modes of traditional industries are changed to form new models, new products, and new services, thus leading to the reconstruction of the industrial ecological system [14,15].

The concept of green development is primarily presented in the studies on the green economy and the ecological economy. Green development projects a new development pattern that achieves sustainable development by protecting the natural environment under scarce resources and certain ecological environments [16]. Furthermore, developed countries are constantly faced with concerns of environmental pollution due to the process of industrialization. Therefore, different countries present different levels of resource endowments and economic development, so the highest critical point of reaching the "Environmental Kuznets" curve would be different [17,18].

The United Nations Environment Programme (UNEP) put forward the Green New Deal in response to the 2008 financial crisis, expecting to promote a new industrial revolution through green investment [19]. Afterward, several developed countries successively improved the energy structure of their innovative green technology, developed green industry, power industry, and green energy by imposing a carbon tax, zero subsidies, implementing a carbon labeling system, and formulating green production norms and industry standards in order to win the market opportunity of green development and achieve economic green transformation [20–22]. The developed countries have transferred polluting industries to developing countries while improving their own environment. Therefore, developing countries are faced with resource constraints and environmental issues in the division of international trade.

Industrial green development reflects the concept of green development from the perspective of the industrial sector. It is an industry geared to realizing cleaner production and green products [23]. Research scholars examine the green development theory, devel-

opment mechanism, approaches, and regional differences based on the regional agriculture, and industrial sectors such as energy, manufacturing, modern services, in addition to the construction industry. The importance of green planning, standardized governance, and policy guidance has been emphasized through the proposed empirical research on green total factor productivity and by developing a comprehensive indicator system for green development [24–26]. Furthermore, industry green high-quality development involves the pursuit of industrial economic benefits while relying on the extensive development model to ensure quality green development [27]. It also focuses on the coordinated development of the industry's economic, ecological, and social benefits to achieve green development, ecological development, and the circular development of the industry [28].

The digital economy and green economy collaborate in terms of different aspects such as technology, industry, and concept to achieve mutual coordination, organic unity, and mutual penetration [29]. One the one hand, digital technology is applied in green technology to improve the level of industrial green technology innovation and the green transformation of the traditional economy, based on the integration of the digital economy in the green economy [30,31]. Digital technology also helps to effectively and rapidly obtain ecological protection information in order to activate digital tracking and detection of R&D, manufacturing, recycling, transportation, and other links of traditional and green industries, and ensure the precise monitoring of energy utilization and carbon emission rates [32–34]. The digital public platform improves the efficiency of resource allocation, optimizes the industrial structure, and reduces the cost of industrial green development [35]. The improvement in digital literacy skills makes it easier for the residents to participate in green consumption and emission reduction actions [36,37]. On the other hand, there is a need to fully implement the concept of green development through green computing and the construction of the digital economy's infrastructure. Such initiatives reduce carbon emissions and accelerate the green digital transformation [38]. To sum up, scholars have undertaken systematic studies on the theoretical basis and mechanism of the digital economy and green economy, but there are few studies on the coupling coordination between the digital economy and industrial green high-quality development. In addition to this, there is also a lack of systematic exploration of the dynamic evolution characteristics, spatial differences, and convergence of the coupling coordinated by both variables. Thus, This paper refers to the relevant theories and thoughts of scholars on the digital economy and green economy, and analyzes the coupling and coordination relationship between the digital economy and high-quality industrial green development from a meso perspective. The possible innovations are as follows: first, we analyze the meaning of industrial green high-quality development, and construct the index system from four aspects: industrial benefit and structure, resource consumption, environmental pollution and control, and industrial recycling development. Secondly, this study adopts the coupling coordination model to estimate and analyze the degree of coupling coordination and spatio-temporal characteristics between the digital economy and industrial green high-quality development in 31 provinces of China for the time-period ranging from 2008 to 2020 in order to comprehensively grasp the association and dynamic evolution trend of coupling coordination. Thirdly, the Dagum Gini coefficient and its decomposition are used in this paper to study the spatial differences and sources of the coupling coordination in China. This research study uses the spatial $\beta$ convergence model to verify the spatial convergence of the coupling coordination degree in China. As a result, practical guidance is proposed in this study for exploring the coupling coordinated path between the digital economy and industrial green high-quality development.

## 2. Index System Construction, Research Methods and Data Sources

*2.1. Index System Construction of Industrial Green High-Quality Development and the Digital Economy*

2.1.1. The System of Industrial Green High-Quality Developmen

Industrial green high-quality development that focuses on the integration of high-quality green aspects is based on two major dimensions. Firstly, the green transformation of traditional industries implies that industrial development no longer depends on high energy consumption, followed by high emissions and high pollution. Alternatively, industries rely on green and low-carbon, technological and scientific innovations to transform industrial development from a factor-driven to an innovation-driven mode; thereby, driving industrial circular development. Secondly, the high-quality development of green industries represents another major aspect. It emphasizes that quality products and greening production areas should continue to reduce the consumption of energy resources and carbon emissions per unit output. Moreover, there is also a need to improve the influence and competitiveness of green products through green scientific and technological innovation. Therefore, industrial green high-quality development contains four core characteristics: "high resource utilization", "high economic efficiency, reasonable structure", "low emissions and low pollution" and "circular development". Consequently, the evaluation system of industrial green high-quality development indicators is composed of four target levels: resource consumption, industrial efficiency and structure, environmental pollution and governance, and industrial circular development.

2.1.2. The System of the Digital Economy

Presently, research institutions have released the digital economy indicator system and measurement methods, such as the International Telecommunication Union, European Union, and Department of Commerce, and the China Center for Information Industry Development [39]. However, there are different standards for these measurement methods. Consistent with the needs of industrial green development, this paper develops indicator systems: the infrastructure of the digital economy, technological innovation, and industrial application of the digital economy. The infrastructure of the digital economy serves as the basis of industrial development, including new infrastructure and traditional infrastructure. Moreover, digital innovation leads to the transformation of industrial structure, including innovation input and innovation output, whereas digital application includes industrial digitalization and digital industrialization (see Table 1).

**Table 1.** The Index System for Industrial Green High-quality Development and the Digital Economy.

| | Target Layer | Criterion Layer | Indicator Layer | Attributes |
|---|---|---|---|---|
| the Index System for Industrial Green High-quality Development | Industry Economy Benefits and Structure | Rationalization of Industrial Structure | New Theil index | − |
| | | Advanced Industrial Structure | The proportion of output value of secondary and tertiary industries in GDP | + |
| | | Labor Productivity | Regional production output/employment (yuan/person) | + |
| | Resource Consumption | Energy Consumption | Energy consumption per ten thousand yuan of regional GDP (standard coal after conversion) | − |
| | | Water Consumption | Per capita water consumption | − |
| | | Land Consumption | Per capita land area | − |
| | Environmental pollution and Control | Environmental Pollution | Wastewater emissions from 10,000 yuan of gross regional product | − |
| | | | Emissions of exhaust gases from 10,000 yuan of gross regional product | − |
| | | | Solid waste emissions from 10,000 yuan of gross regional product | − |
| | | Environmental Control | The proportion of industrial pollution control investment in industrial value added (%) | − |
| | | | Comprehensive utilization rate of solid waste (%) | + |

**Table 1.** *Cont.*

| Target Layer | Criterion Layer | Indicator Layer | Attributes |
|---|---|---|---|
| | Green energy | Proportion of clean energy | + |
| | Green industry | Proportion of main business income of high-tech industry in GDP | + |
| Circular development of industry | Green investment | Proportion of expenditure on energy conservation and environmental protection in local public financial expenditure | + |
| | Green jobs | Proportion of employees in high-tech industries in total employment | + |
| | Traditional Infrastructure | Number of internet broadband port access (million households) | + |
| Digital Basic Condition | | Internet penetration rate (%) | + |
| | New infrastructure | Phone penetration rate (units per 100 people) | + |
| | | Length of toll cable | + |
| | Innovation input | Proportion of R&D investment in high-tech industries in total R&D investment | + |
| Digital innovation | Innovation output | Sales Revenue of New Products | + |
| The Index System for Digital Economy | | Turnover of technology market | + |
| | Digital Industrialization | Share of ICT employment in total regional employment (%) Proportion of ICT employment in total regional employment (%) | + |
| Application of Digital Industry | | Proportion of software Revenue in gross regional product | + |
| | Industrial Digitization | Total telecom business | + |
| | | Number of websites per 100 enterprises (number) | + |
| | | E-commerce sales | + |
| | | Digital financial inclusion index | + |

The stated indicators are weighted through the entropy method. The proposed method is a commonly used research method of scholars, therefore, this paper will not outline a detailed description. Besides, the calculation method can refer to the description of Jun W. et al. [40].

*2.2. Research Methods*

(1) Coupling coordination model: The dynamic relationship of development and coordination between systems can be described by the coupling coordination degree [41]. This model has been widely used in the fields of economics and geography. This paper measures the coupling coordination degree between the digital economy and industrial green high-quality development by adopting the research technique proposed by Hao L. et al. [42]. Firstly, it requires computing the relative development degree ($E$) of industrial green high-quality development and the digital economy. $E = \frac{U_g}{U_d}$, $U_g$ and $U_d$ represent the level of industrial green high-quality development and the digital economy, respectively. In the case $0 < E \leq 0.8$, it indicates that industrial green high-quality development is lagging behind; thereby restricting the development of the digital economy. Alternatively, when $0.8 < E < 1.2$, it means that both the digital economy and industrial green high-quality development are developing simultaneously. In addition to this, when $E \geq 1.2$, it shows that the digital economy is lagging behind, thus restricting industrial green high-quality development [43].

Secondly, There is a need to estimate the coupling degree $C$. involves calculating the coupling degree $C$. involves calculating the coupling degree $C$. $C = \left\{ \frac{U_g U_d}{\left[ (U_g + U_d)/2 \right]^2} \right\}^{\frac{1}{2}}$.

The coupling degree between $U_g$ and $U_d$ is higher when $C$ is larger. This index focuses on the consistency between the two subsystems, therefore, it cannot reflect the dynamic coordination degree. Finally, there is a need to estimate the coupling coordination degree, $T$. $T = a_1 U_g + a_2 U_d$, $D = \sqrt{C \times T}$. $T$ denotes the index of coupling coordination between the digital economy and industrial green high-quality development, while $a_1$ and $a_2$ represent the importance degree of the two subsystems. It is commonly believed that both are equally

important, $a_1 = a_2 = 0.5$. Lastly, $D$ stands for coupling coordination. The larger its value, the stronger the coupling coordination of the two subsystems.

(2) Dagum Gini coefficient and its decomposition: The Gini coefficient proposed by Dagum (1997) is applied to test the regional difference of the coupling coordination in order to effectively avoid the sources of data overlapping and regional difference. Furthermore, the overall Gini coefficient is categorized into three parts: intra-regional difference and its contribution ($G_w$), inter-regional difference and its contribution ($G_{nb}$) and hypervariable density and its contribution ($G_t$) [44]. The calculation formula is the mathematical expression for estimation of the Gini coefficient, as follows:

$$G = \frac{\sum_{p=1}^{k} \sum_{q=1}^{k} \sum_{i=1}^{N_p} \sum_{r=1}^{N_q} \left| D_{pi} - D_{qr} \right|}{2N^2 \overline{D}} \tag{1}$$

$i$ and $r$ represent the serial number of provinces in the different regions whereas $N$ denotes the total provinces (31). Similarly, $k$ shows the serial number of the region (total 4). Meanwhile, $N_p$ and $N_q$ indicate total provinces in the region $p$ and the region $q$. Lastly, $D_{pi}$ and $D_{qr}$ represent the coupling coordination level of $i$ province in region $p$ and $r$ province in region $q$, respectively. $\overline{D}$ stands for the mean value of the coupling coordination degree between the digital economy and industrial green high-quality development.

Gini coefficient and the contribution rate of intra-regional difference in the $p$ region are as below:

$$Gpp = \frac{\sum_{i=1}^{N_p} \sum_{r=1}^{N_q} \left| D_{pi} - D_{qr} \right|}{2N^2 \overline{D}}, \quad G_w = \sum_{p=1}^{N} G_{pp} h_p s_p$$

The Gini coefficient value and the contribution rate between the $p$ region and $q$ region are as follows:

$$Gpq = \frac{\sum_{i=1}^{N_p} \sum_{r=1}^{N_q} \left| D_{pi} - D_{qr} \right|}{N_p N_q (\overline{D}_p - \overline{D}_q)}, \quad G_{nb} = \sum_{p=2}^{N} \sum_{q=1}^{p-1} G_{pq} (h_p s_q + h_q s_p) F_{pq}$$

The contribution of hypervariable density is:

$$G_t = \sum_{p=2}^{N} \sum_{q=1}^{p-1} G_{pq} (h_p s_q + h_q s_p)(1 - F_{pq})$$

Wherein, $h_p = \frac{N_p}{N}$, $s_p = \frac{N_p \overline{D}_p}{N \overline{D}}$; $F_{pq}$ captures the relative influence of the coupling coordination degree between the digital economy and industrial green high-quality development in the $p$ and $q$ region, which can be expressed as follows: $F_{pq} = \frac{(d_{pq} - h_{pq})}{(d_{pq} + h_{pq})}$, $d_{pq}$ and $h_{pq}$ represent the mathematical expected values of the sum of all samples of $(D_{pi} - D_{qr}) > 0$ and $(D_{pi} - D_{qr}) < 0$ in regions $p$ and $q$, respectively.

(3) B Convergence model: B Convergence is classified into absolute $\beta$ Convergence and conditional $\beta$ Convergence, developed by Elhorst (2010). This paper applies the spatial Dubin model to verify the $\beta$ Convergence process of the coupling coordination between the digital economy and industrial green high-quality development [45]. The proposed model is as follows:

$$\begin{aligned} D'_{i,t+1} &= \ln(\frac{D_{i,t+1}}{D_{it}}) = \alpha + \beta_1 \ln D_{it} + \beta_1 Controls_{it} + \rho \omega_{ij} D'_{i,t+1} + \gamma_1 \omega_{ij} \ln D_{it} + \gamma_2 \omega_{ij} Controls_{it} \\ &\quad + \mu_i + \eta_t + \varepsilon_{it} \end{aligned} \tag{2}$$

In model (2), $D'$ represents the growth rate of coupling coordination degree, while, $i$ stands for the province, and $t$ shows the time. Additionally, $\ln D_{it}$ and $\ln D_{i,t+1}$ are the evaluation index of coupling coordination degree in $t$ and $t + 1$, with $\alpha$ as a constant term. Besides this, $\beta$ represents the convergence coefficient. The significance level test is passed when $\beta < 0$, indicating that the initial investigated value is negatively correlated with its growth rate

and demonstrates the characteristic of convergence, with convergence speed as $\nu = -\ln(1+\beta)/T$. Otherwise, it indicates divergence. Furthermore, $\rho$ and $\gamma$ show the spatial effect coefficient, with $w$ as the spatial weight, and $j$ as the $j$-th variable in the control variables. Subsequently, $\mu_i$ reflects the individual effect, with $\eta_t$ as the time effect, and $\varepsilon_{it}$ as the random error term. Model (2) represents the spatial error model (SEM) when the spatial coefficients are all 0. Conversely, model (2) shows the spatial lag model (SLM) when $\gamma_1 = \gamma_2 = 0$. Lastly, the Controls represent the control variables with conditional $\beta$ convergence, including economic development level (PGDP), finance (Fin), technological innovation (Tec), local fiscal expenditure (GE) and openness (Open). The controls are replaced by GDP per capita, the proportion of total institutional deposits and loans in GDP, the number of patent grants, the proportion of total local financial budget expenditure in GDP, and proportion of total import and export in GDP. The logarithmic transformation is carried out during the analysis since the PGDP and Tec variables show an exponential change trend with time.

### 2.3. Data Source

This study measures the level of the digital economy and industrial green high-quality development in 31 provinces of China from 2008 to 2020. There are two reasons: firstly, data from Taiwan, Hong Kong and Macao are difficult to obtain; secondly, many data statistics methods of China's digital economy before and after 2008 are inconsistent. The original data used are from the *China Statistical Yearbook*, the statistical yearbooks of provinces, the *China Environmental Statistical Yearbook*, the *China Energy Statistical Yearbook*, the *China Science and Technology Statistical Yearbook*, and the Ai Media Data Center.

The new Theil index: The new Theil index constructed by Gan et al. (2011) is used in this study to measure the rationalization of industrial structure [46]. The mathematical expression for the new Theil index is as $NTL = \sum_{i=1}^{n} \left( \frac{Y_i}{Y} \right) Ln \left[ \frac{(Y_i/L_i)}{(Y/L)} \right]$, where $n$ refers to the number of industries, $Y$ represents $GDP$, $Y_i$ stands for the $i$ industrial output, $L$ denotes the total employment, and $L_i$ means the number of $i$ industrial employees.

## 3. An Empirical Analysis of the Coupling Coordination between the Digital Economy and Industrial Green High-Quality Development

### 3.1. Spatio-Temporal Characteristics and Spatial Differences of the Coupling Coordination

### 3.1.1. Spatio-Temporal Characteristics

The overall evolution of the coupling coordination degree between the digital economy and industrial green high-quality development in Table 2 demonstrates the following characteristics: firstly, the digital economy lags behind industrial green high-quality development in 2008 and 2009. Moreover, the digital economy develops synchronously with industrial green high-quality development during 2010–2020. Secondly, the degree of coupling coordination exceeds 0.9 during the investigation, indicating that the degree of coupling coordination is in a highly coupled state and reflects a strong interaction. Thirdly, the coupling coordination degree shows an increasing trend. This means that China's extensive economic growth model has weak innovation awareness and serious environmental pollution at the stage of declining coupling coordination. Once China's economy entered a stage of high-quality development, it thoroughly carried out the five development ideas of "innovation, coordination, green, openness, and sharing". Afterward, it also paid attention to industrial green development, which made the steady growth of the coupling coordination degree between the digital economy and industrial green high-quality development. However, the coupling coordination degree displays a slow growth rate, thus confirming that there is still a need to continuously concern the coupling coordination.

**Table 2.** Time Trend of the Coupling Coordination Degree.

| Year | $U_d$ | $U_g$ | E | C | D |
|------|-------|-------|------|-------|-------|
| 2008 | 0.614 | 0.766 | 1.248 | 0.993 | 0.828 |
| 2009 | 0.628 | 0.773 | 1.231 | 0.995 | 0.835 |
| 2010 | 0.667 | 0.746 | 1.118 | 0.998 | 0.840 |
| 2011 | 0.754 | 0.737 | 0.977 | 0.999 | 0.863 |
| 2012 | 0.760 | 0.713 | 0.938 | 0.999 | 0.858 |
| 2013 | 0.764 | 0.704 | 0.921 | 0.999 | 0.856 |
| 2014 | 0.765 | 0.701 | 0.916 | 0.999 | 0.857 |
| 2015 | 0.808 | 0.730 | 0.903 | 0.999 | 0.876 |
| 2016 | 0.815 | 0.751 | 0.921 | 0.999 | 0.884 |
| 2017 | 0.821 | 0.778 | 0.947 | 0.999 | 0.894 |
| 2018 | 0.825 | 0.789 | 0.956 | 0.999 | 0.898 |
| 2019 | 0.828 | 0.790 | 0.954 | 0.999 | 0.900 |
| 2020 | 0.830 | 0.792 | 0.954 | 0.999 | 0.900 |

Figure 1 illustrates that the degree of coupling coordination in the eastern region is higher than that of other regions. In addition, the degree of coupling coordination in the central regions, northeast regions, and western regions is lower than that of the whole country. The coupling coordination degree in the central region demonstrates the fastest growth rate due to the rapid improvement of the digital economy and upgrading of the

green industry after 2017. Since the coupling environment and economic foundation of the digital economy and industrial green development in the eastern region are the most mature, the siphon effect on technology, finance, and other resources is also relatively strong. Besides this, the development foundation of other regions is weak. The central region has been strongly supported by regional policies in recent years; therefore, the marginal effect of the coupling coordination degree of the two subsystems is the most significant.

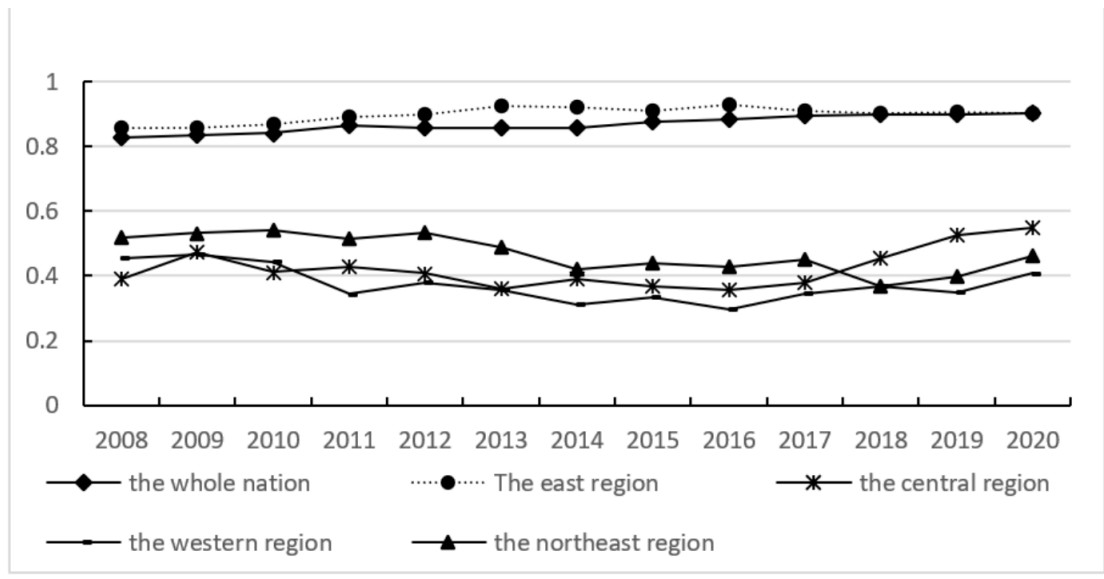

**Figure 1.** Spatial Trend of the Coupling Coordination Degree.

### 3.1.2. Regional Differences

Dagum Gini coefficient and its decomposition are used to reveal the regional differences and sources of the coupling coordination degree between the digital economy and industrial green high-quality development.

(1) Overall difference analysis: Figure 2 confirms that the overall difference of the whole nation is commonly on the decline, thereby indicating that the coupling coordination degree difference is shrinking nationwide with the continuous support for industrial green development, but the overall spatial difference is still relatively prominent.

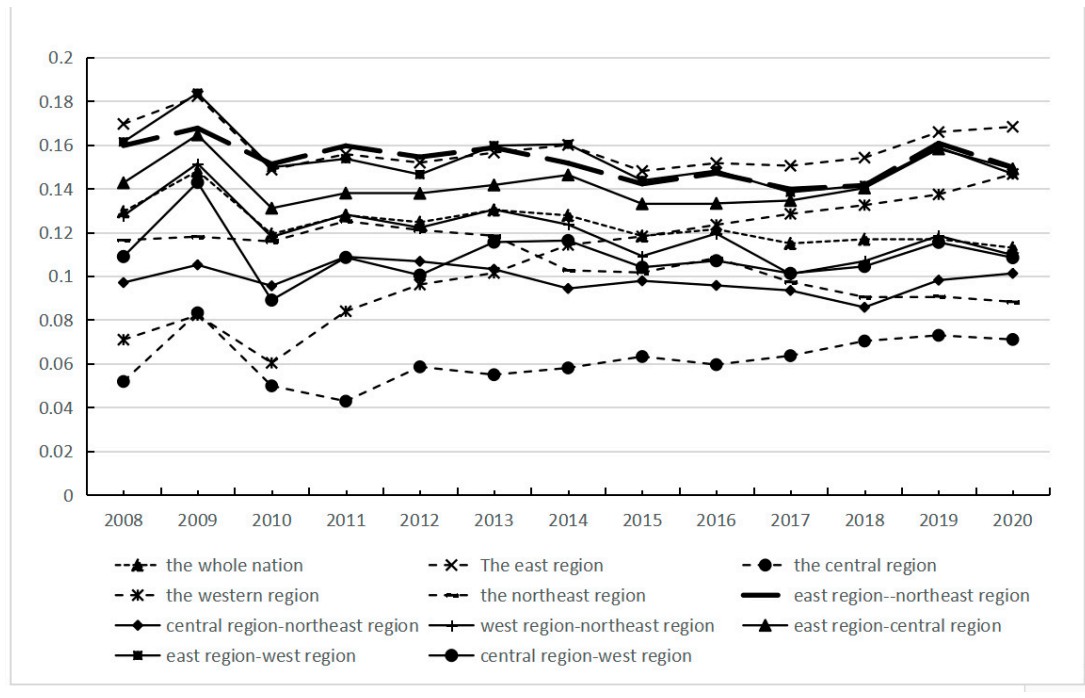

**Figure 2.** Regional Differences of the Coupling Coordination Degree.

(2) Intra-regional differences: There are obvious spatial differences in the four regions. Firstly, the Intra-regional difference in the eastern region is the largest, with a Gini coefficient of around 0.16. Beijing, Shanghai, Zhejiang, Guangdong, and Jiangsu display strong economic strength, as these cities have substantially promoted industrial green development and a digital economy. Besides, the degree of coupling and coordination is significantly higher for these regions than that of Tianjin, Hebei, Fujian, Shandong, and Hainan. The intra-regional difference exhibited an expanding trend during the period from 2008 to 2020. Secondly, the intra-regional difference in the western region is smaller, although its growth rate is higher. For instance, the digital transformation of green industries such as Chongqing, Sichuan, and Shaanxi is high. In particular, Chongqing and Sichuan have formed a pole of China's digital economy. The coupling coordination's growth rate in both regions is far higher than that of other provinces in the western region. Thirdly, the intra-regional difference in the central region is the lowest with an upward trend. Fourthly, the Gini coefficient value in the Northeast is below 0.1 with a declining trend which indicates that since the development of all provinces in the Northeast is balanced, therefore the gap is small. The major reason behind this is the similarity of industrial development of all provinces in Northeast China.

(3) Inter-regional differences: The east-west region represents the largest regional difference, with an average of 0.153, followed by the east-northeast region and the east-central region. The Gini coefficient of the west-northeast, middle-west, and middle-northeast regions is relatively small, with mean values of 0.12, 0.11 and 0.1, respectively. This indicates that inter-regional differences in China are majorly driven by the spatial differences in the east-west, east-northeast, and east-Northeast, although the western, central, and northeastern regions have advantages in terms of resources and energy endowments compared to the eastern region. Meanwhile, the supporting conditions in several aspects are slightly insufficient, thereby leading to the relative backwardness of the digital economy and industrial green development. Since there are more similarities in the development of the western, central, and northeast regions, there are therefore small differences between the regions.

(4) Sources and contributions of differences: Table 3 shows that the contribution rate of the three sources of the difference is highly stable. Hypervariable density serves as the major source of spatial differences. The intra-regional contribution rate fluctuates at 22%, while the inter-regional contribution rate is the lowest, displaying a downward trend. Conversely, the contribution of hypervariable density exhibits an upward trend, with the contribution rate up to 64.69%. This means that the coupling coordination degree of the four regions has a certain intersection. In addition, the industrial development of certain provinces in different regions is similar. As a result, a province with a lower coupling coordination degree in the higher rank region may be lower than a province with a higher value in the lower rank region.

**Table 3.** Sources and Contribution Results of Regional Differences.

| Year | Intra-Regional Differences | | Inter-Regional Differences | | Hypervariable Density | |
|------|-------|-------------------|-------|-------------------|-------|-------------------|
| | Value | Contribution Rate | Value | Contribution Rate | Value | Contribution Rate |
| 2008 | 0.0295 | 22.77% | 0.0264 | 20.42% | 0.0735 | 56.81% |
| 2009 | 0.0337 | 22.79% | 0.0304 | 20.52% | 0.0839 | 56.69% |
| 2010 | 0.0269 | 22.55% | 0.0324 | 27.18% | 0.0599 | 50.27% |
| 2011 | 0.028 | 21.89% | 0.0326 | 25.49% | 0.0672 | 52.62% |
| 2012 | 0.0282 | 22.60% | 0.0294 | 23.60% | 0.067 | 53.79% |
| 2013 | 0.0289 | 22.18% | 0.0339 | 26.04% | 0.0674 | 51.78% |
| 2014 | 0.0285 | 22.30% | 0.0281 | 22.03% | 0.0711 | 55.67% |
| 2015 | 0.0267 | 22.59% | 0.0253 | 21.36% | 0.0663 | 56.05% |
| 2016 | 0.0274 | 22.57% | 0.0269 | 22.16% | 0.0671 | 55.27% |
| 2017 | 0.026 | 22.62% | 0.0211 | 18.35% | 0.0678 | 59.04% |
| 2018 | 0.0265 | 22.72% | 0.017 | 14.53% | 0.0733 | 62.75% |
| 2019 | 0.0274 | 22.47% | 0.0134 | 10.26% | 0.0761 | 67.27% |
| 2020 | 0.026 | 22.62% | 0.0157 | 12.69% | 0.0712 | 64.69% |

### 3.2. β Convergence Test of the Coupling Coordination

#### 3.2.1. Spatial Correlation Test

Stata12.0 software is used in this study to conduct the Moran's I test on the coupling coordination degree between the digital economy and industrial green high-quality development. The coupling coordination degree is significantly positive for the period ranging from 2008 to 2020, indicating that the neighboring regions affect the level of coupling coordination in a region (see Table 4).

**Table 4.** Moran's I Index of Coupling coordination.

| Year | Moran's I | z | p |
|------|-----------|------|-------|
| 2008 | 0.137 | 2.881 | 0.004 |
| 2009 | 0.176 | 3.53 | 0 |
| 2010 | 0.135 | 2.859 | 0.004 |
| 2011 | 0.206 | 4.033 | 0 |
| 2012 | 0.201 | 3.953 | 0 |
| 2013 | 0.192 | 3.812 | 0 |

**Table 4.** *Cont.*

| Year | Moran's I | z | p |
|---|---|---|---|
| 2014 | 0.197 | 3.891 | 0 |
| 2015 | 0.185 | 3.671 | 0 |
| 2016 | 0.186 | 3.691 | 0 |
| 2017 | 0.171 | 3.448 | 0.001 |
| 2018 | 0.153 | 3.139 | 0.002 |
| 2019 | 0.186 | 3.673 | 0 |
| 2020 | 0.195 | 3.743 | 0.001 |

### 3.2.2. $\beta$ Convergence Test

(1) The convergence test of absolute $\beta$: The spatial Dubin model is finally used through the model adaptation test. The result of the Hausman test is $p < 0.05$, thus leading to the rejection of the random effect's null hypothesis. This means that the spatial Dubin model with the two-way fixed effect can be adopted in this paper to test the $\beta$ convergence of the coupling coordination between the digital economy and industrial green high-quality development.

It is evident from Table 5 that the $\beta$ coefficients are significantly negative. This implies that the coupling coordination degree has absolute $\beta$ convergence with control variables (economic development, technological innovation, finance, openness, and local fiscal expenditure) being equally similar. Moreover, the eastern region reflects the fastest convergence rate, the second in the northeastern, the third in the central, and the slowest in the western region, which is below the national average. This means that the intra-regional effect of the eastern region is the most prominent in promoting coupling coordination without considering differences in other influencing factors.

**Table 5.** $\beta$ Convergence Test of Coupling.

| | Variable | The Whole Nation | The Eastern Region | The Central Region | The Western Region | The Northeastern Region |
|---|---|---|---|---|---|---|
| Absolute $\beta$ Convergence test | D | −0.401 *** | −0.461 *** | −0.70 *** | −0.39 *** | −0.556 *** |
| | W × D | 0.11 | 0.060 | 0.57 ** | −0.05 | −0.06 |
| | $\rho$ | 0.082 | −0.170 | −0.23 | 0.02 | 0.15 * |
| | $v$ | 0.039 | 0.047 | 0.093 | 0.038 | 0.063 |
| | $R^2$ | 0.79 | 0.95 | 0.71 | 0.50 | 0.64 |
| | LogL | 833.71 | 311.58 | 202.73 | 287.97 | 316.73 |
| Conditional $\beta$ Convergence test | D | −0.16 *** | −0.26 *** | −0.48 *** | −0.27 *** | −0.33 *** |
| | $d(PGDP)$ | 0.01 * | 0.03 | 0.05 | 0.012 * | 0.015 |
| | $d(Tec)$ | 0.01 * | 0.06 * | 0.04 *** | 0.02 * | 0.01 ** |
| | $Open$ | 0.05 | 0.08 ** | 0.03 | −0.2 | −0.1 |
| | $GE$ | 0.001 *** | 0.001 ** | 0.01 | 0.001 | 0.001 |
| | $Fin$ | 0.02 ** | 0.05 ** | 0.06 | 0.01 *** | 0.01 |
| | $v$ | 0.013 | 0.026 | 0.050 | 0.024 | 0.031 |
| | Space lag term of control variable | yes | yes | yes | yes | yes |
| | $\rho$ | 0.14 | 0.03 | 0.38 | 0.25 | 0.14 |
| | $R^2$ | 0.43 | 0.47 | 0.59 | 0.2 | 0.37 |
| | LogL | 693.83 | 288.64 | 173.79 | 252.11 | 126.73 |

Note: ***, **, * are Significant at 1%, 5% and 10% Levels Respectively.

(2) Convergence test of conditional $\beta$: The Hausman test result $p > 0.05$, supports the null hypothesis of the random effect model. Therefore, this study performs the conditional $\beta$ convergence test using a random effect. Table 4 contains the influence of conditional $\beta$ convergence of economic development, finance, technological innovation, local fiscal expenditure, and openness. The estimated regression results are as follows: firstly, the coefficients of conditional $\beta$ in the whole nation and the four regions are all significantly negative. This reflects that there exists a convergence trend of conditional $\beta$ in these regions. The central region displays the fastest convergence speed, with a value of 0.05, while the western region demonstrates the slowest convergence speed. Furthermore, the speed of all regions is higher than that of the nation. The control variables in the central region have the most significant coordination benefits. that the central region has the most obvious coupling coordination effect of the control variables. Secondly, the factors of regions are not only quite different, but the influence degree of the $\beta$ convergence process is also different. (1) Consequently, economic development exerts a positive impact on the coupling coordination between the digital economy and industrial green high-quality development. This means that economic development provides material support to improve coupling coordination. However, there is only significant convergence of the coupling coordination between the country and the western region. Therefore, it is essential to focus on supporting the economic development of the western region while adopting the digital economy as a driving force of development and exploring the turning point of industrial green and high-quality development. (2) Technological innovation shows a significant positive influence on the coupling coordination degree. In addition, it promotes the coupling coordination level to converge at a high intensity, but the steady-state values are different, thereby indicating that the technological innovation promotes the coupling coordination degree between the digital economy and industrial green high-quality development, but the impact degree is

different in different regions. (3) The openness significantly promotes the coupling coordination degree in the eastern region to convergence at a high level, and the impact on the western and northeast regions is negative. This indicates that openness helps improve the coupling coordination level in the eastern region, and there is a need to further expand the opening degree of other regions. (4) Lastly, finance significantly supports the coupling coordination degree of the whole country, The convergence speed of the coupling coordination degree of the whole country, the eastern region and the western region has accelerated by the financial development. Additionally, the impact coefficient on the central and northeast region is positive but insignificant. This infers that the utilization efficiency and reasonable allocation of the financial resources affect the level of coupling coordination. Thus, sufficient attention should be paid to the imbalance of excessive and low utilization of financial resources in the eastern region and the lack of financial resources in the western underdeveloped provinces. In addition to this, the local fiscal expenditure significantly enhances the coupling coordination degree to high intensity in eastern China. However, the proposed effect is positive but insignificant in the western, central, and northeast regions. This reflects that the local fiscal expenditure could partly improve the coupling coordination between the digital economy and industrial green high-quality development, owing to the possible crowding-out effect of local fiscal expenditure on private credit, which hinders the effective flow of capital elements across different regions and industries, thus impeding industrial green high-quality development.

## 4. Conclusions and Suggestions

This paper conducts an empirical test on the degree of coupling coordination between the digital economy and industrial green high-quality development based on the spatial data of 31 provinces in China for a time period ranging from 2008 to 2020. The major conclusions of this study are as follows:

(1) The overall coupling coordination level between the digital economy and industrial green high-quality development represents a high and steady-upward trend in China. Furthermore, the degree of coupling coordination in the four regions is significantly different, as the eastern region shows a higher degree than other regions while the central region observes the fastest growth rate in China.

(2) There is a gradual decline in the overall spatial difference of the coupling coordination between the digital economy and green high-quality industrial development in China, thereby indicating a weakened overall imbalance in the country. Similarly, the intra-regional difference in the eastern region is the largest, with a gradual-rising trend. The growth rate in the western region is the highest, while that in the central region is the lowest. Parallel to this, the intra-regional difference in the northeast region is narrowing over time. These differences between east-west, east-northeast, and east-central regions are large, whereas the Gini coefficient of west-northeast, central-west, and central-northeast regions is relatively small. Subsequently, the hypervariable density serves as the major contribution to spatial difference.

(3) There is a stable positive spatial correlation of coupling coordination degree between the digital economy and industrial green high-quality development. Furthermore, the coupling coordination degree of the above regions displays a convergence trend of absolute $\beta$, with the eastern region showing the fastest convergence speed when the influencing factors of economic development, openness, technological innovation, finance, and local fiscal expenditure are the same. Contrarily, there is the convergence of conditional $\beta$ in regions, in the case of the regional differences in the influencing factors. As a result, the resulting spatial effects are different.

Based on the study conclusions, the authors propose the following suggestions:

Firstly, this study fully realizes that the coupling coordination level of the digital economy and industrial green high-quality development is constantly improving. However, the growth rate of coupling and coordination is still slow. Therefore, there is a need for a "top-down + bottom-up" strategic plan. It is essential to guide green policy and ideology from the perspective of the top-down approach. In addition, more social capital is required for investment in the digital economy and industrial green upgrading, owing to the leverage effect formed by the guidance of government funds. Conversely, there is a need to cultivate digital and green literacy from the perspective of the bottom-up approach. People consciously carry out green travel and green low-carbon consumption, thereby improving the market demand for green products and services, and promoting the coupling coordination between the digital economy and industrial green high-quality development.

Secondly, there is a need to be aware of the spatial unbalanced distribution of the coupling coordination. The eastern region plays a leading role and narrows the gap within the region and narrows down the gap in coupling coordination within the region. The central region increases investment in scientific research, consequently exploring the cultivation and retention channels of compound talents who possess "digital + green + industry knowledge" [34]. The northeast region deepens the market reform, eliminates low-end and outdated production capacity, improves the business environment, and optimizes and upgrades its industrial structure with the support of digital technology innovation. Moreover, most provinces in the western region are on the verge of imbalance. These provinces continue to expand openness, fully stimulate the green vitality of industries, promote the construction of new infrastructure, and accelerate the optimization, upgrading, and green transformation of traditional industries. Meanwhile, the eastern cities and provinces should develop a long-term cooperation mechanism with other regions, strengthen the free flow of factors, jointly build and share a digital platform, expand the application scenarios of industrial green development, and improve the resource reallocation between regions and industries.

Thirdly, the degree of coupling coordination converges under the influence of control variables, consistent with the characteristics of $\beta$ convergence over the period of time. Therefore, it is compulsory to implement precise policies in accordance with the influencing factors in different regions. (1) Economic development serves as the material basis for the coupling coordination and fully taps the driving force and potential of economic

development in the western region. (2) The spatial spillover effect of technological innovation should be expanded in the eastern region. In the same vein, market-oriented reform must be deepened, and innovation vitality should be stimulated in the central, western, and northeast regions. (3) There is also a need for a greater opening to the outside world, to accelerate the construction of a smooth domestic and international circulation market, and to increase the market green demand. (4) Furthermore, a special fund must be set up to support the coordinated development between the digital economy and industrial green high-quality development in central, western, and northeast regions. The creation of a favorable policy environment provides fair and equal investment opportunities for private capital. Fifthly, financial resources should be guided to other regions through policies. (5) The western region needs to continue to improve green finance and digital inclusive finance in order to support more capital flows to the green industrial sectors through financial means such as preferential interest rates.

## 5. Discussions and Limitations

This paper conducts an empirical test about the space-time characteristics, regional differences and convergence on the degree of coupling coordination between the digital economy and industrial green high-quality development. It is concluded that the overall coupling coordination level is higher, and the eastern region is the highest as Hu S etc. [29]. Although Zheng measured green economy from a macro perspective, the conclusions are consistent, because industry is the main carrier of economic development. Zheng X. et al. [32] noted the regional differences between the digital economy and the green economy. This paper divides the region into east, middle, west and northeast regions, and analyzes the inter-regional and intra-regional differences and their contributions. The hypervariable density serves as the major contribution to spatial difference, which expands and supplements Zha J. and Chen L.'s conclusion [38] and verifies Qian L. etc.'s theoretical idea [33] that the degree of overlapping influence between regions is higher. Hu S. etc. [29] analyzed the convergence rate of each region with the control variables and concluded that the convergence rate is the fastest in the western region, which is different from the conclusion of the faster convergence rate in the eastern region of this paper. The possible reasons for this difference are the different control variables added and the different regional divisions. This paper divides the relatively backward provinces of Liaoning, Heilongjiang and Jilin in the eastern region into the northeast region, which affects the conclusions that follow the analysis.

The research samples were 31 provinces and cities in China from 2008 to 2020, which may have some influence on the results due to the short time of sample selection because some data statistics methods for measuring the digital economy before and after 2008 are different. At the same time, the number of control variables may also limit research conclusions. We will increase the number of control variables and constantly improve the research results.

This study makes the government and enterprises realize that it is necessary to pay attention to the coordinated development between the digital economy and industrial green high quality development. Each region attaches the problem of spatial imbalance through the spatial differences and should not only play the leading role of developed cities, but also make precise decisions according to local conditions and characteristics. For example, the eastern region needs to stimulate innovation vitality, and the western region needs to increase financial support.

**Author Contributions:** Formal analysis, T.G.; Writing—original draft, L.L.; Writing—review & editing, H.W. All authors have read and agreed to the published version of the manuscript.

**Funding:** This research received no external funding.

**Institutional Review Board Statement:** Not applicable.

**Informed Consent Statement:** Not applicable.

**Data Availability Statement:** The data are in my article that can be downloaded at: https://doi.org/10.6084/m9.figshare.21670016.

**Conflicts of Interest:** The authors declare that they have no conflict of interest.

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
