# Peer review of "The Coupling Coordination between Digital Economy and Industrial Green High-Quality Development: Spatio-Temporal Characteristics, Differences and Convergence"

_sustainability, doi:10.3390/su142316260_

Round 1
Reviewer 1 Report
The paper's composition is coherent; the structure is logical and meets the goal of the paper. The title "The Coupling Coordination Between Digital Economy and Industrial Green High-Quality Development Spatio-Temporal Characteristics, Differences and Convergence" puts well the paper's objective; it is clear and expresses the issue being assessed very well. The abstract is formulated adequately along with the true picture of the paper. All the tools and methods the author uses are reasonable and well described and adequately fit the problem being assessed to give the reliable results. Conclusions are related to the results presented before reflecting the assessed issue at a professional level. All the tables are complete and understandable. Authors use enough calculations, charts, and tables featuring a great deal of data being processed hence adding a higher added value to the paper. However major revision would suffice to get the manuscript published in the journal. It is recommended that the authors make a relatively major revision, and the specific amendments to the text are as follows:
- In introduction to underline the added value, novelty, and ways of application of the research results would be recommended.
- The goal explicitly stated within the Introduction clearly expressing the main problem and purpose of the paper and author's intention being assessed and discussed within the paper along with its clear and unambiguous formulation are required to be proposed.
- Please specify the originality and relevance of the topic in the field of Industry 4.0 and world and national economies and industry, especially the synergy and parallels between the digital economy and green economy.
- The Discussion is recommended to be set aside from the section Findings/Empirical Analysis and Conclusion. Some kind of polemic discourse comparing the research outcomes with the literature overview part would be beneficial to be involved in Discussion.
- I recommend adding to the Conclusion section authors’ further research directions within this explored issue along with a brief research limitation.
- In Conclusion section there could be provided a final statement reflecting the assessed issue along with the way how the research results could be implemented in the practice bringing up any benefits and added value.
Author Response
Thank you for your constructive comments on our paper. We have learned much from it. We have made corresponding changes to our paper according to your comments.

Reviewer 2 Report
Dear authors,
On the scientific layer, I have no comments or objections. A well-written article, interesting issues, good scientific argumentation, the content of the article with appropriate literature sources. At the same time, I would like to emphasize that I do not feel competent to assess the performed statistical analysis and its correctness. On the other hand, the presented conclusions seem to be correct and convincing.
In addition, I think that the scientific value of the article, taking into account its aim & purpose, would certainly increase significantly thanks to the presentation of the results obtained from bibliometric software - for example VOSviewer. The introduction of this extra content is not mandatory of course, but very much appreciated. Alternatively, using VOSViewer and presenting the results could be an idea for another article in the series.
For "technical" remarks:
1. It is worth considering a better location of figure 1 (at present it is mentioned in line 282, while the drawing itself is only in line 334 and is preceded by a lot of text - a bit confusing / illegible)
2. Ummm "3.2. βConvergence Test of the Coupling Coordination" section is missing... or there are mistakes in the numbering of individual parts of the article (line 349 = 3.2 / line 350 = 3.3.1. / line 357 = 3.2.2
3. Missing - Supplementary Materials: The following supporting information can 492 be downloaded at: ()line 492 & 493) and Data Availability Statement: The data are in Supplementary Material (line 498)
Author Response

(The authors gave the same response as above.)

Round 2
Reviewer 1 Report
The revised paper titled “The Coupling Coordination Between Digital Economy and Industrial Green High-quality Development - Spatio-temporal Characteristics, Differences and Convergence “ intended to be published in Sustainability Journal meets all the requirements for a professional scientific journal. All the significant comments, recommendations and remarks of reviewers have been incorporated into the manuscript in a proper way giving the paper higher added value and professional features.